# Upscale Synthesis of Magnetic Mesoporous Silica Nanoparticles and Application to Metal Ion Separation: Nanosafety Evaluation

**DOI:** 10.3390/nano13243155

**Published:** 2023-12-16

**Authors:** Mathilde Ménard, Lamiaa M. A. Ali, Ani Vardanyan, Clarence Charnay, Laurence Raehm, Frédérique Cunin, Aurélie Bessière, Erwan Oliviero, Theodossis A. Theodossiou, Gulaim A. Seisenbaeva, Magali Gary-Bobo, Jean-Olivier Durand

**Affiliations:** 1ICGM, Univ Montpellier, CNRS, ENSCM, 34193 Montpellier, France; mathilde.menard@umontpellier.fr (M.M.); clarence.charnay@umontpellier.fr (C.C.); laurence.raehm@umontpellier.fr (L.R.); frederique.cunin@enscm.fr (F.C.); aurelie.bessiere@umontpellier.fr (A.B.); erwan.oliviero@umontpellier.fr (E.O.); 2IBMM, Univ Montpellier, CNRS, ENSCM, 34193 Montpellier, France; miss_limo@yahoo.com (L.M.A.A.); magali.gary-bobo@inserm.fr (M.G.-B.); 3Department of Biochemistry, Medical Research Institute, University of Alexandria, Alexandria 21561, Egypt; 4Department of Molecular Sciences, Swedish University of Agricultural Sciences, 750 07 Uppsala, Sweden; ani.vardanyan@slu.se (A.V.); gulaim.seisenbaeva@slu.se (G.A.S.); 5Department of Radiation Biology, Institute for Cancer Research, Radium Hospital, Oslo University Hospital, Montebello, 0379 Oslo, Norway; theodossis.theodossiou@rr-research.no

**Keywords:** magnetic mesoporous nanoparticles, upscaled synthesis, DTPA, metal ion separation, toxicity

## Abstract

The synthesis of core–shell magnetic mesoporous nanoparticles (MMSNs) through a phase transfer process is usually performed at the 100–250 mg scale. At the gram scale, nanoparticles without cores or with multicore systems are observed. Iron oxide core nanoparticles (IO) were synthesized through a thermal decomposition procedure of α-FeO(OH) in oleic acid. A phase transfer from chloroform to water was then performed in order to wrap the IO nanoparticles with a mesoporous silica shell through the sol–gel procedure. MMSNs were then functionalized with DTPA (diethylenetriaminepentacetic acid) and used for the separation of metal ions. Their toxicity was evaluated. The phase transfer procedure was crucial to obtaining MMSNs on a large scale. Three synthesis parameters were rigorously controlled: temperature, time and glassware. The homogeneous dispersion of MMSNs on the gram scale was successfully obtained. After functionalization with DTPA, the MMSN-DTPAs were shown to have a strong affinity for Ni ions. Furthermore, toxicity was evaluated in cells, zebrafish and seahorse cell metabolic assays, and the nanoparticles were found to be nontoxic. We developed a method of preparing MMSNs at the gram scale. After functionalization with DTPA, the nanoparticles were efficient in metal ion removal and separation; furthermore, no toxicity was noticed up to 125 µg mL^−1^ in zebrafish.

## 1. Introduction

The use of core–shell magnetic mesoporous silica nanoparticles (MMSNs) has grown a lot over the last decade. Many different applications have been described, such as their use as theranostic agents for magnetic resonance imaging and drug delivery, for cancer applications [1,2,3], sensor agents for sensitive detections [4,5], or extracting agents for the removal of heavy metal ions [6,7,8], rare earth elements [9] or actinides [10]. Another recent study highlights the potential ability of MMSNs to remove iron from biological media [11]. The field of MMSNs and their applications have been comprehensively discussed and reviewed [12,13,14,15,16,17,18,19,20,21,22]. Two groups independently reported the first preparation of MMSNs in 2008 [23,24]. Zink’s group encapsulated iron oxide nanoparticle cores with 20 nm diameters in MSNs of 100–200 nm diameters. The preparation first consisted of the thermal decomposition of iron–oleate complexes at 320 °C in octadecene–oleic acid to produce the nanoparticle core. Then, the iron-oxide nanocrystals coated with oleic acid were transferred into an aqueous solution with cetyltrimethylammonium bromide (CTAB) through the dispersion of the nanocrystals in chloroform and the evaporation of the volatiles. The subsequent hydrolysis–polycondensation of tetraethoxysilane in the presence of CTAB led to MMSNs. Hyeon’s group essentially used the same procedure for the synthesis of monodisperse MMSNs of different diameters (45–105 nm) with cores of 15 nm, and 22 nm cores were also used. Note that MMSN synthesis is not an easy task and is usually performed at the 150–250 mg scale in order to obtain homogeneous monodisperse, well-dispersed core–shell nanoparticles. The preparation often leads to a mixture of nanoparticles with multiple Fe_3_O_4_ cores per particle or mesoporous silica nanoparticles without cores, particularly when upscaling is carried out. In the course of our work on MMSNs for metal removal, we present here a detailed procedure in order to synthesize monodisperse MMSNs with 100 nm diameters and an 18 nm magnetic core diameter at the gram scale. MMSNs were then functionalized with diethyl triamino pentacetic acid (DTPA) for the extraction and separation of Ni(II), Co(II), Sm(III) and Nd(III) ions. The nontoxicity of MMSNs was evaluated in cells, zebrafish embryos, and with seahorse metabolic assays.

## 2. Materials and Methods

### 2.1. Materials

The reagents are of 99% purity (not technical grade). Iron oxide α-FeO(OH) (CAS 20344-49-4, Sigma Aldrich, Saint-Quentin-Fallavier, France), oleic acid (CAS 112-80-1, Sigma Aldrich Saint-Quentin-Fallavier, France), *n*-docosane (CAS 629-97-0, Acros, Illkirch, France), oleylamine (CAS 112-90-3, Acros, Illkirch, France), tetraethyl orthosilicate TEOS (CAS 78-10-4, Sigma Aldrich Saint-Quentin-Fallavier, France), cetyltrimethylammonium bromide CTAB (CAS 57-09-0, Sigma Aldrich Saint-Quentin-Fallavier, France), pentane (CAS 109-66-0, Sigma Aldrich, Saint-Quentin-Fallavier, France), chloroform (CAS 67-66-3, Sigma Aldrich, Saint-Quentin-Fallavier, France), trifluoroacetic acid (CAS 76-05-1, Fluorochem, Hadfield, UK), triethylamine (CAS 121-44-8, Merck, Saint-Quentin-Fallavier, France), dichloromethane (CAS 76-09-2, Carlo Erba, Val de Reuil, France), ethyl acetate (CAS 141-78-6, VWR, Rosny-sous-Bois, France), ethanol and acetone (Honeywell, Vitrolles, France).

The *FTIR* spectra were recorded in the 4000–400 cm^−1^ range using 32 scans at a nominal resolution of 4 cm^−1^ using a Perkin Elmer 100 FT spectrophotometer (Villebon-sur-Yvette, France) equipped with an ATR unit. The *TEM* images were recorded with a JEOL 1200 EXII microscope (JEOL Europe SAS, Croissy Sur Seine, France). For the purpose of the TEM analysis, the sample particles were dispersed in ethanol and then deposited onto copper grids covered with porous carbon films. *TGA analyses* were performed with a thermal analyzer STA 409 Luxx^®^ (Netzsch, Selb, Germany) in the range 25–800 °C and at a heating speed of 5 °C/min. DLS was performed with the use of a Cordouan Technologies DL 135 (Pessac, France). Zeta potentials were measured by a Malvern Nanoseries zetasizer, (Orsay, France), and the SEM-EDS technique was performed using a Flex-SEM 1000 II scanning electron microscope (Krefeld Germany).

Metabolic assays, OCR and ECAR measurements, were performed on a Seahorse XFe96 Analyzer (Agilent Technologies, Santa Clara, CA, USA).

### 2.2. Procedures

#### 2.2.1. Synthesis of Magnetic Iron Oxide Nanoparticles (MIONs)

The IO nanoparticles were synthesized through thermal decomposition according to a previous reported protocol [25]. Briefly, 0.18 g of hydrated iron oxide (III) was mixed with 5 g of docosane and 3.2 g of oleic acid. The mixture was initially stirred under vacuum conditions for 30 min and then placed under an Ar flow for another 30 min. Then, the reaction occurred when heating at 340 °C for 1 h 30. After the completion of the reaction, the IO nanoparticles were washed and recovered by three successive centrifugations at 20 krpm for 10 min and redispersion by ultrasound for a few minutes: (1) by adding 15 mL of pentane and 30 mL of a mixture of ether:ethanol (2:1, *v*:*v*) to the as-synthesized product, (2) by adding 2 mL of pentane and 30 mL of a mixture of ether:ethanol (2:1, *v*:*v*) to the collected black product, and (3) by adding 30 mL of a mixture of ether:ethanol (1:1, *v*:*v*) to the collected black product. Finally the IO nanoparticles were stored in 15 mL of chloroform and stabilized by adding 200 µL of oleylamine; 120 mg of IO nanoparticles were obtained.

#### 2.2.2. Synthesis of MMSNs

The scale-up protocol to synthesize 1 g of MMSNs was designed by adapting a previously reported protocol that usually provided between 150 and 200 mg [1]. In the optimized process, 2 g of CTAB (a reduction of 25% compared to ref [1]) was first dissolved in 250 mL of distilled water for 1 h at 70 °C under stirring in a 1 L flat- and large-bottom flask (Erlenmeyer or beaker) to enhance the liquid/air interface. The surfactant solution was then cooled down to 40 °C, and 5 mL of the previously synthesized IO nanoparticles dispersed in chloroform were added. The emulsion was first subjected to vigorous stirring without heating for 30 min, and then the temperature was raised to 70 °C for an additional 40 min of stirring to ensure the transfer of the IO nanoparticles from an organic to aqueous phase. Once the phase transfer of the IO nanoparticles was completed, 300 mL of distilled water, slightly basified by 300 µL of 2 M NaOH, was added to the solution under stirring. When the solution reached 60 °C, 7.5 mL of TEOS, followed by 12.5 mL of EtOAc, was quickly added to the mixture. Then, the condensation reaction was conducted for 3 h at 70 °C under stirring. The final MMSNs were recovered by centrifugation (5 to 10 min at 20 krpm), concentrated to 100 mL and washed two times: twice with water and twice with EtOH at 95%. After each centrifugation step, the MMSNs were redispersed in the appropriate washing solvent under sonication for a few minutes. To extract the CTAB, the MMSNs were soaked twice in a solution of ammonium nitrate (6 g/L in EtOH 90%) and placed under stirring at 70 °C for 2 h. After each extraction, the washing steps by centrifugation described above in water and ethanol were performed.

Finally, the MMSNs (1 g obtained, with a yield around 62% based on the TEOS consumption) were redispersed in 25 mL of 96% EtOH.

#### 2.2.3. Functionalization of MMSNs with DTPA

In a typical procedure, 2 mmol of DTPA per g of MMSNs was added to a suspension of MMSNs at 10 mg/mL in EtOH.

Briefly, 217 mg of diethylene triamine pentacetic anhydride, 135 mg of APTES, and 61 mg of triethylamine were mixed in 40 mL of DMF under ultrasound for 30 min. The mixture was then placed under stirring for 3 h at room temperature. Then, 30 mL of a suspension of MMSNs at a concentration of 10 mg/mL in EtOH were added. The solution was treated with an ultrasound for 1 h and then stirred at 80 °C under reflux for 12 h. Finally, the grafted nanoparticles, MMSN-DTPAs, were washed by centrifugation once with EtOH, twice with water and once with acetone, and then redispersed in EtOH.

#### 2.2.4. Isotherm Experiments of Ni^2+^, Co^2+^, Sm^3+^ and Nd^3+^ Adsorption

For the isotherm experiments, stock solutions of metal cations (50 mM) were prepared using nitrate salts, and the final concentrations were adjusted by dilution with milli-q water (varying between 0.5 and 25 mM). Samples (10 mg) of magnetic nanoparticles (MMSNs) were mixed with 10 mL of a metal cation solution in plastic tubes of 50 mL and placed on a shaker for 24 h. After each experiment, the MMSNs were centrifuged (7000 g) for 10 min, and an aliquot (1 mL) was separated to determine the metal concentration in the remaining solution. The samples were first diluted 5–10 times (depending on the initial metal concentration) and titrated afterwards with EDTA using xylenol orange as an indicator. For each sample, the titrations were repeated 3 times, and the average concentration was calculated. 

The uptake of metal cations by the magnetic nanoparticles was calculated according to the equation:Up = (C_o_ − C_e_) × V/m
where C_o_ is the initial metal concentration, which was also measured by titration; C_e_ is the equilibrium metal concentration; V is the metal solution volume, which was kept constant; and m is the NPs weight.

For the kinetic tests, the magnetic nanoadsorbents (20 mg) were mixed with 20 mL of metal cation solutions (with a 10 mM initial metal concentration), and the uptake was measured after set time intervals. For that purpose, an aliquot of 1 mL was separated and diluted 10 times, and the remaining metal concentration was evaluated by titration with EDTA and xylenol orange.

For testing the selectivity, solutions of Ni:Nd, Co:Sm and Ni:Co in 1:1 ratios were used. First, 10 mg of the adsorbent and 10 mL of the test solutions in 50 mL plastic tubes were placed on a shaker for 24 h. The particles were separated from metal solution by centrifugation (7000 g) and dried under a nitrogen atmosphere. They were subsequently analyzed by SEM-EDS to determine the metal ratios.

#### 2.2.5. Cell Culture

Adult human dermal fibroblasts (HDF) were purchased from Lifeline Cell Technology (Frederick, MD, USA). Cells were maintained in Roswell Park Memorial Institute (RPMI) 1640 medium (Gibco, Life technologies limited, Paisley, UK) supplemented with 10% fetal bovine serum (FBS) and penicillin/streptomycin (Gibco, Life technologies limited, Paisley, UK). Cells were allowed to grow in a 5% CO_2_ humidified atmosphere at 37 °C using a HERACell incubator (Fisher Scientific SAS, Illkirch-Graffenstaden, France). 

#### 2.2.6. Cytotoxicity Study

Cells were seeded in a 96-well plate (Corning, NY, USA) at a density of 5000 cells *per* well. Twenty four hours after seeding, the cells were treated with different concentrations (ranging from 0 to 250 µg mL^−1^) of MMSNs or MMSN-DTPAs, and then the cells were incubated for 72 h. After the incubation period, the cells were processed for measuring their metabolic activity using the colorimetric MTT (3-(4,5-dimethylthiazolyl-2)-2, 5-diphenyltetrazolium bromide) assay (M5655-1G, Sigma-Aldrich, Saint-Quentin-Fallavier, France). Accordingly, the cells were incubated with an MTT solution in a culture medium at a concentration of 0.5 mg mL^−1^ for 4 h, and then the medium was aspirated, and the formazan crystals were dissolved using a mixture of DMSO/ethanol (1:1, *v*:*v*) (Sigma-Aldrich, Saint-Quentin-Fallavier, France) with agitation for 20 min. The absorbance was read at 540 nm using a Multiskan SkyHigh Microplate Spectrophotometer (Fisher Scientific SAS, France). The percentages of viable cells were calculated according to the following equation: %(viable cells) = Ab_test_/Ab_control_ × 100. The experiment was repeated 3 times. The dose–response curve was plotted as the log of the nanoparticles concentration in µg mL^−1^ versus the percentage of viable cells using GraphPad Prism 5.0 software (San Diego, CA, USA). The mean lethal concentration (LC_50_) was determined at 50% cell death. 

#### 2.2.7. Toxicity Study in Zebrafish (Danio Rerio) Embryos

Wild-type zebrafish embryos purchased from the Zebrafish International Resource Center (Eugene, OR, USA) were used. These embryos were raised to adulthood in a circulating aquarium system at 28 °C, an 80% humidity and a 14 h light/10 h dark cycle in the lab’s facilities of Molecular Mechanisms in Neurodegenerative Dementias (MMDN), Inserm U1198, Montpellier University, Montpellier. On the day of the experiment, males and females were mated to produce fertilized embryos that were collected and maintained at 28 °C. Five hours post-fertilization (hpf), the embryos were examined under a Loupe Olympus MVX10 stereomicroscope in order to choose the normally developed ones for the experiment. Embryos were placed in a 12-well plate (Corning, NY, USA) (20 embryos in each well) and exposed to 4 mL of water containing different concentrations (10, 25, 125 and 250 µg mL^−1^) of MMSNs or MMSN-DTPAs. The mortality and viability (hatched or chorionated) were observed under the Olympus MVX10 stereomicroscope (Rungis, France) at different time points, which ended at 80 hpf. The experiment was repeated 3 times. The statistical difference between the embryo death in the control and tested groups was analyzed by Student’s t-test using GraphPad Prism 5.0 software (San Diego, CA, USA). The level of significance was defined as * *p* < 0.05 and ** *p* < 0.005.

Experiments with zebrafish embryos until 96 hpf are considered in vitro studies according to the EU Directive 2010/63/EU on the protection of animals used for scientific purposes.

#### 2.2.8. Metabolic Assay

SKOV3 and T47D cells were purchased from the ATCC biobank.

For the metabolic assays, we used the XFe96 metabolic analyzer (Seahorse, Agilent Technologies, Santa Clara, CA, USA). Thirty thousand cells were plated in Seahorse XFe96 plates and left overnight to attach. One hour prior to the assay, the cell medium was changed to an XF RPMI 1640 unbuffered medium (Agilent) supplemented with 10 mM of glucose, 2 mM of L-glutamine and 2 mM of sodium pyruvate and incubated at 37 °C and 0% CO_2_ for 1 h. The MMSNs were then injected in situ into the respective cell groups, and then all cells were subjected to a “mitostess” test consisting of (i) oligomycin (2 μΜ) for the inhibition of the F_1_F_0_-ATPase and to show the proportion of respiration corresponding to ATP production, (ii) FCCP (1 μM) to uncouple the cell mitochondria and reveal the maximal respiratory capacity and (iii) Rotentone/Antimycin A (2/2 μΜ) to completely inhibit the electron transport. The above were administered as sequential injections (3 measurement cycles each), following the initial basal measurements (medium-only; 4 measurement cycles). In all cases, both the oxygen consumption rates (OCRs) and extracellular acidification rates (ECARs) were determined. The background OCRs and ECAR were obtained from wells without cells (medium only) and subtracted automatically by the XFe96 2.6 software.

The metabolic profile of the iron oxide core, the MMSNs, with and without DTPA, was studied in two human cell lines, SKOV3 ovarian adenocarcinoma and T47D breast ductal carcinoma. The cells were seeded in Agilent Seahorse XF96 cell culture 96-well plates at 30,000 cells per well and left overnight to attach in a humidified 5% CO_2_ atmosphere. Two cell groups were incubated for 4 h with two concentrations of MMSNs (±DTPA), 5 and 50 μg/mL, respectively, while the control cells were kept in media. The first 3 hours of incubation were carried out in complete cell media and in a humidified 5% CO_2_ atmosphere. During the last hour, however, the cells were incubated in an unbuffered seahorse RPMI medium (pH 7.4) without FBS and in the absence of CO_2_. Four basal oxygen consumption rate (OCR) and extracellular acidification rate (ECAR) measurements were conducted in all cell groups before the injection of 1 μM of oligomycin A, an inhibitor of the F_0_-F_1_ ATPase. Three measurements were conducted following the injection to reveal the OCR required for ATP synthesis. The second injection that followed was 1 μM of FCCP, a protonophore which uncouples electron transport from oxidative phosphorylation, to show the maximal respiratory capacity. Following three measurements under the FCCP addition, the cells were injected with a cocktail of 1 μM of rotenone and antimycin A, specific inhibitors of the quinone-reducing centers of complex I and III, respectively. The final three measurements under rotenone and antimycin A represent the background measurements of no mitochondrial respiration since the two drugs completely inhibit electron transport.

## 3. Results and Discussion

### 3.1. Preparation of MMSNs and MMSN-DTPAs

The principle of the synthesis and functionalization of MMSNs is presented in Figure 1.

The synthesis of core–shell nanoparticles made of an iron oxide core and a porous silica shell was deeply investigated and described this past decade. Indeed, researchers were able to precisely tune the size of the shell by varying the Fe/surfactant ratio [24]. Nevertheless, to the best of our knowledge, the syntheses of homogeneous core–shell nanoparticles never exceed the 100–200 mg scale. Thus, the scale-up of these objects to the gram scale remains challenging. We first investigated the scale-up experiment by homogeneously increasing all quantities (TEOS, surfactant, iron oxide core, solvents) to reach the gram scale. This method only led to a non-negligible amount of unwanted “no-core” porous silica particles mixed with the wanted core–shell particles; see Figure 2a. A size analysis made by measuring more than 400 nanoparticles from TEM images showed that the number of “no-core” nanoparticles is close to the number of core–shell NPs (48% vs. 52%, respectively, Figure 2b). 

As an excess of surfactant, which can act as a soft template to initiate the sol–gel reaction, could lead to the presence of “no-core” porous silica particles, the CTAB amount was decreased from 25% compared to the 100 mg protocol. This simple protocol modification allowed a gram of monodispersed core–shell NPs to be obtained, but unfortunately this was not always reproducible, which implies that other key parameters had to be considered. Indeed, although the presence of “no-core” NPs was not noticed, “multiple-core” NPs were observed in some experiments (Figure 3). 

Two key parameters which can lead to this result were identified. First: the stability of the core material. Indeed, iron oxide (IO) cores which are not perfectly dispersed and stabilized in chloroform will aggregate and form “multiple core”–shell structures. These dispersion issues were often due to extensive washing steps after thermal decomposition that desorb oleic acid from the IO surface. This can be easily verified by DLS measurements on IO cores before the sol–gel reaction step or by observing the sedimentation of the sample when stored. The second parameter is harder to control: the phase transfer of the IO cores from chloroform to an aqueous solution. This phase transfer, which is widely described in the literature [26], is based on the formation of water-soluble nanomicelles driven by the hydrophobic Van der Waals interactions between the alkyl chains of the surfactant and the oil-stabilizing ligand, forming an interdigitated bilayer structure around the IO core. The complete removal of chloroform during the phase transfer is also necessary to obtain water-stable nanomicelles without clustering. Thus, controlling these key synthesis steps is important to succeeding in the scale-up. Three synthesis parameters were controlled: temperature, time and glassware. Before the addition of IO NPs, the solution containing dissolved CTAB had to be cooled down to 40 °C to avoid the instant evaporation of chloroform. To ensure a good oil/water emulsion, the mixture was placed under stirring without heating for 30 min. Then, to secure the phase transfer of IO NPs through complete chloroform evaporation and proper Van der Waals interactions between oleic acid and CTAB, the temperature was increased to 70 °C, and the mixture was kept at this temperature under stirring for an additional 40 min. Finally, to achieve a better stirring to foster oil/water emulsion and a higher liquid/air interface to facilitate chloroform evaporation, the syntheses were performed in large-bottom beakers or in Erlenmeyer flasks. The rest of the protocol, which consisted of the sol–gel reaction (dilution in water, the addition of a base, TEOS and EtOAc, and aging at 70 °C for 3 h), was kept identical, as previously developed. Following this method, homogeneous monodispersed core–shell nanoparticles were obtained; see Figure 4. These NPs were then used for the surface grafting of DTPA.

To assess the success of DTPA in grafting onto MMSNs, comparative to FTIR, TGA and Zeta analyses were performed. 

The apparition on the FTIR spectra of ν_N-C_ at 1395 cm^−1^, ν HO-_C=O_ at 1700 cm^−1^ and ν NH-_C=O_ at 1650 cm^−1^ demonstrated the grafting of DTPA (Figure 5). The different characteristic bands, ν Si-O (1100 cm^−1^) and ν C-H (2920, 2855 cm^−1^), showed that the mesoporous silica shell was not damaged. The zeta potential was measured in water as a function of pH, which allowed the isoelectric point of the MMSNs to be determined at pH = 3.5, which was in agreement with the deprotonation of Si-OH. The isoelectric point with the MMSN-DTPAs was lower (pH = 3), which is in agreement with the deprotonation of carboxylic acid groups (pKa (DTPA) = 1.68, 2.1, 2.6, 4.15) [27] (Figure 6). Finally, the TGA analyses showed, for MMSNs, a loss of adsorbed water from 0 °C to 200 °C, and then the residual organics (surfactants) were calcined from 200 to 600 °C, leading to a mass loss of 15%. For MMSN-DTPAs, a final weight loss difference of 20.5% with MMSNs was observed, which corresponds to a grafting of 0.6 mmol of DTPA per gram of MMSNs (Figure 7).

### 3.2. Adsorption of Heavy Metals Using MMSNs and MMSN-DTPAs

Rare earth elements (REEs) are key components of green energy and high-tech growth industries. For example, there are about 320 kg of REEs in the magnet of a wind turbine. Separating elements from outdated systems is therefore crucial for recycling REEs. Raw MMSNs showed a considerably lower capacity for both heavy metals (1.00 mmol/g) and appreciably less for the REEs (0.88 and 0.85 mmol/g for Nd and Sm, respectively), whereas with DTPA, the surface modification almost doubled the adsorption capacity (Table 1). 

The adsorption isotherms of the metal cations were best fit with Langmuir curves, having higher correlation coefficients compared to the Freundlich model (Figure 8, Table 2).

Many adsorbents have previously been tested for REE and LTM (late transition metal) removal from aqueous solutions. Table 3 summarizes the Langmuir monolayer capacity of various results with silica-based and other adsorbent materials. It can be noted that MMSNs present a competitive performance for the binding of selected metals with high adsorption capacities.

The kinetic test results showed a fast uptake in the first 1–2 h of the metal’s interaction with MMSN-DTPAs, reaching over 70% of the total capacity. A slower uptake was observed after these 2 h, and equilibrium was reached after 5–6 h (Figure 9).

Selectivity tests revealed a considerable difference in the metal uptake in pairs, showing the appreciable preference of MMSN-DTPAs for Ni as opposed to Nd and Ni rather than Co, while lower selectivity was observed in the Co-Sm pair in favor of Sm (see Table 4).

### 3.3. Nanosafety

First, the nanosafety of MMSNs and MMSN-DTPAs was investigated by in vitro cell viability studies on healthy human dermal fibroblasts. After 3 days of incubation with varying concentrations of MMSNs or MMSN-DTPAs, a similar cytotoxic response was observed for both nanoparticles (Figure 10). The cell viability results showed classical sigmoidal dose-response curves when plotted versus the logarithmic function of the nanoparticle concentration (Figure 10). The calculation of the LC_50_ values for both nanoparticles yielded the same value of 32 µg mL^−1^. These results indicated that the surface modification does not have a crucial impact on the nanoparticles’ cytotoxic behavior in these experimental conditions. Further, Baber et al. reported the protective effect of the silica coat against the acidic erosion of the magnetic core, which results in reduced cytotoxic behavior [33].

The toxic effect of MMSNs or MMSN-DTPAs on zebrafish embryos’ development was evaluated, as shown in Figure 11. Embryos at 5 h post fertilization (hpf) were exposed to water containing different concentrations of nanoparticles, and the mortality and viability of the embryos were observed until 80 hpf. The results showed that no significant toxicity (death) was observed for the MMSNs, even at 250 µg mL^−1^, in comparison with the control, whereas the toxicity was significant with the DTPA coating at this concentration, reaching 65 ± 14% of mortality at 80 hpf (Figure 11). Nevertheless, the MMSN-DTPAs did not have a significant toxicity at lower concentrations.

We then examined the metabolism of SKOV3 and T47D cells through the seahorse metabolic assay. The results of the seahorse metabolic measurements are presented in Figure 12. The oxygen consumption rate (OCR) measurements appear in Figure 12a, while the extracellular acidification rate (ECAR) measurements are shown in Figure 12b. From the data presented in Figure 12a, we can observe that the T47D cells are less respiratory than the SKOV3 cells. In addition, the T47d cells exhibited equal basal and maximal respiration rates, while for the SKOV3 cells, their maximal respiration (carbonyl cyanide 4-(trifluoromethoxy)phenylhydrazone (FCCP)-uncoupled mitochondria) was approximately double their basal respiration. In the T47D cells, there was a marginally significant reduction in both the basal and maximal respiration upon incubation with either 5 or 50 μg/mL of MMSNs, with no notable concentration dependence. Nevertheless, following incubation with MMSN-DTPAs, there was a significant reduction in the respiration rates (both basal and maximal) by approximately 50%. Again, this drop seemed to be similar for the 5 and 50 μg/mL MMSN-DTPA concentrations. In the case of the SKOV3 cells, the basal and maximal respiration rates (as seen through the OCR) did not seem to change after incubation with 5 μg/mL of MMSNs, but there was an increase in the basal respiration for incubation with 50 μg/mL of the order of ~25%, while a similar response was observed following incubation with either 5 or 50 μg/mL of MMSN-DTPAs. A similar trend was followed for the maximal respiration of the SKOV3 cells; however, in this case, the increase was of the order of 40–50%.

Regarding ECAR, the two cell lines exhibited similar basal glycolysis rates; however, upon the addition of oligomycin, the T47D cells did not significantly increase their glycolysis rate, while in the case of the SKOV3 cells, the glycolysis rate almost doubled upon the oligomycin stimulus. Notably, oligomycin inhibits respiratory ATP production, and hence cells increase their glycolytic activity to compensate for the ATP loss. With respect to the T47D cells, there was no notable change in their ECAR and hence glycolysis rate upon incubation with the nanoparticles with or without DTPA, regardless of the incubation concentration. The SKOV3 cells, however, reacted by approximately doubling their glycolysis rates following incubation with 5 μg/mL of either the MMSNs or MMSN-DTPAs. The incubation with 50 μg/mL of either nanoparticle moiety, however, caused a reduction in the glycolysis rate by 25%. The maximal glycolytic capacity of the SKOV3 cells also followed a similar trend. 

In general, SKOV3 cells are more metabolically active than T47D cells and are also more reactive to stress stimuli, both with regard to raising their respiration and glycolysis rates. In addition, the two cell lines responded differently to incubation with the nanoparticles under investigation. The T47D respiration was impaired following incubation with the MMSN-DTPAs. This impairment was not concentration-dependent, at least in the range of 5–50 μg/mL, and the cell line was not able to compensate this by increasing the glycolysis rate since the T47D cells seem to perform basal glycolysis at their highest possible capacity. The respiratory and glycolytic increases in SKOV3 cells following their incubation with the nanoparticles indicate an increased requirement for energy following the insult, which, however, did not reach their maximal glycolytic capacity (as registered by the addition of oligomycin). However, in the case of the incubation of the SKOV3 cells with 50 μg/mL of nanoparticles, regardless of nanoparticle moiety, the glycolytic rate dropped back down in comparison to that with 5 μg/mL of nanoparticles, which indicates a probable impairment of the glycolytic machinery at these higher concentrations. This could also account for the slight trend of SKOV3 cell’s increasing respiratory rate upon incubation with 50 μg/mL of nanoparticles (even though this increase is not statistically significant). 

In conclusion, while the MMSNs were well tolerated at the two incubation concentrations employed, any adverse effects that were registered, particularly at the highest incubation concentrations, were cell-type dependent. 

## 4. Conclusions

The preparation of the MMSNs though the thermal decomposition of α-FeO(OH) was performed at the gram scale. The key parameters were the dispersion of the IO nanoparticle cores and the phase transfer procedure from CHCl_3_ to water, which necessitated larger neck glassware in order to properly evaporate CHCl_3_. After functionalization with DTPA, the MMSN-DTPAs were efficient in adsorbing Ni ions, with a very good affinity and selectivity for Ni compared to Co and Nd. Compared to other procedures in the literature, the developed technology was competitive. The nanosafety of the MMSNs and MMSN-DTPAs was examined in cells, and the toxicity of the MMSN-DTPAs was low. No toxicity was noticed in the zebrafish model, showing that the mesoporous silica shell has a protective effect on MMSNs. The metabolic experiments in cells showed that the MMSNs and MMSN-DTPAs are well tolerated. 

## Figures and Tables

**Figure 1 nanomaterials-13-03155-f001:**
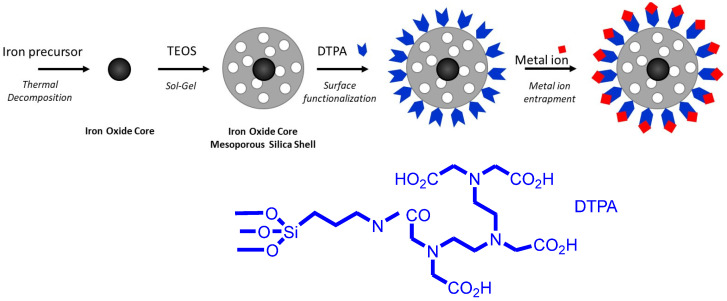
Preparation of MMSNs functionalized with DTPA.

**Figure 2 nanomaterials-13-03155-f002:**
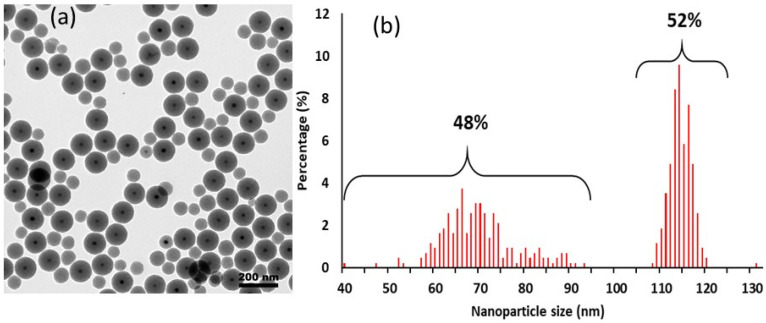
Morphological effect of an excess of CTAB on the scale-up protocol: (**a**) Bright-field TEM image (scale bar: 200 nm) and (**b**) size analysis (on 428 NPs from TEM images).

**Figure 3 nanomaterials-13-03155-f003:**
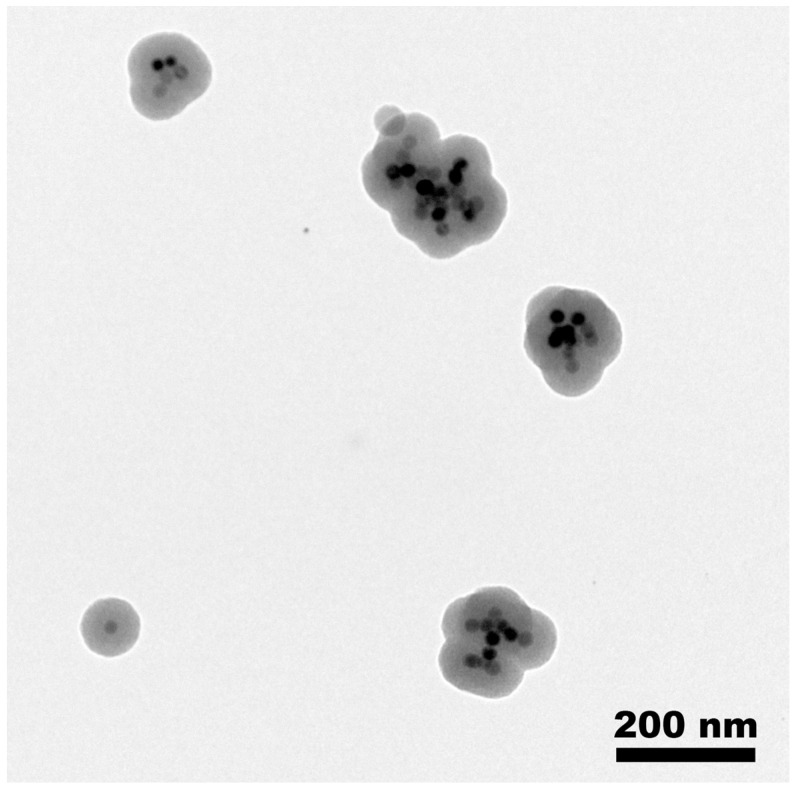
Bright-field TEM image showing the morphological effect of a fail in the phase transfer process (scale bar: 200 nm).

**Figure 4 nanomaterials-13-03155-f004:**
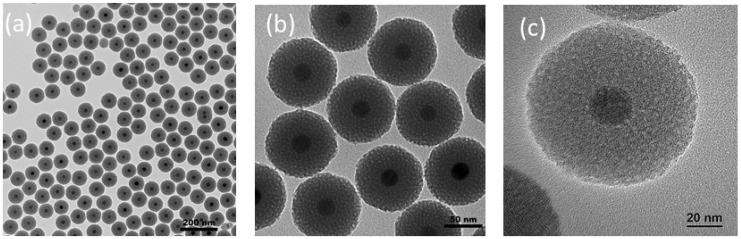
Bright-field TEM images of MMSNs obtained through the 1 g-scale protocol; (**a**,**b**) 120 kV TEM; (**c**) 200 kV high-resolution TEM (HRTEM) showing the porosity of MMSNs.

**Figure 5 nanomaterials-13-03155-f005:**
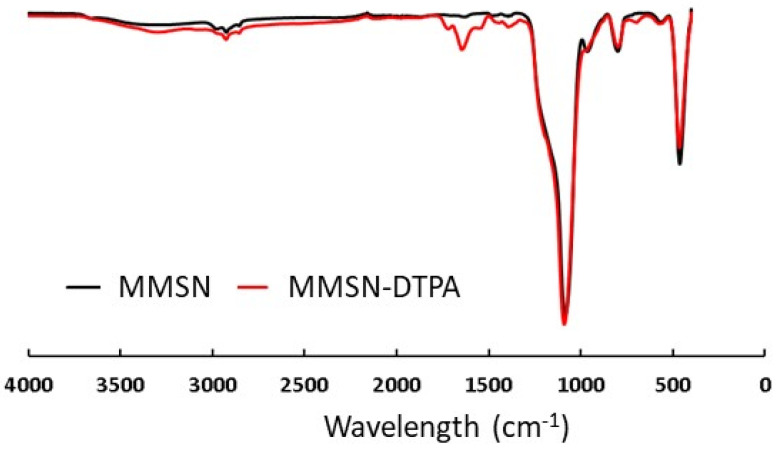
FTIR of MMSNs and MMSN-DTPAs showing functionalization.

**Figure 6 nanomaterials-13-03155-f006:**
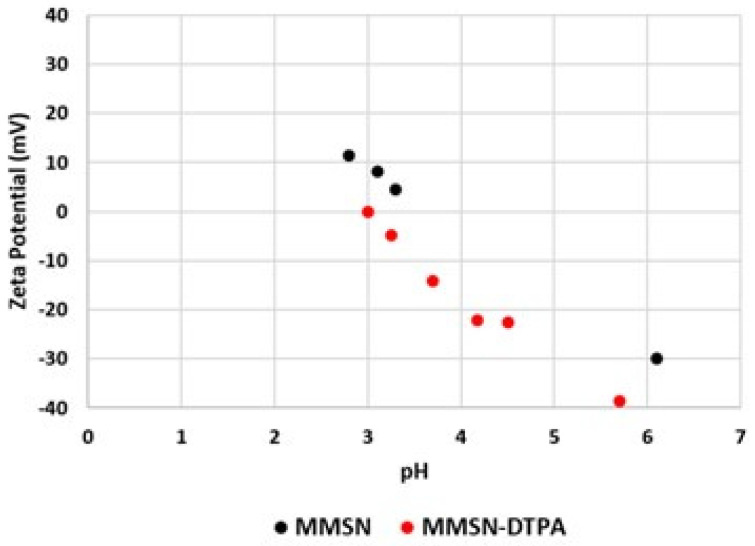
Zeta potential of MMSNs and MMSN-DTPAs.

**Figure 7 nanomaterials-13-03155-f007:**
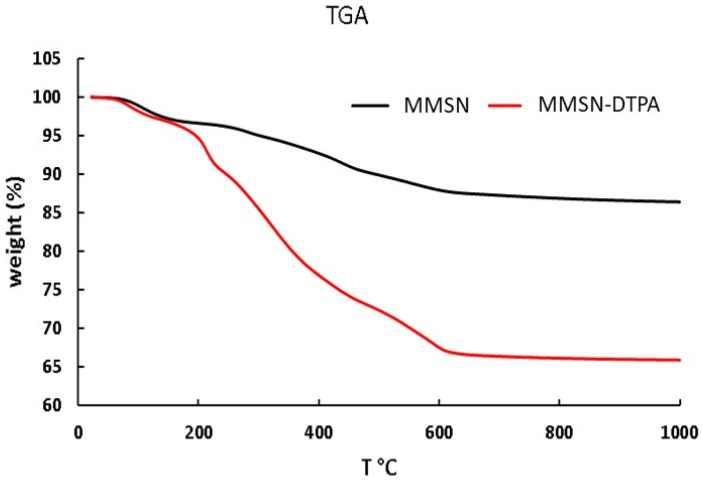
TGA of MMSNs and MMSN-DTPAs showing the functionalization.

**Figure 8 nanomaterials-13-03155-f008:**
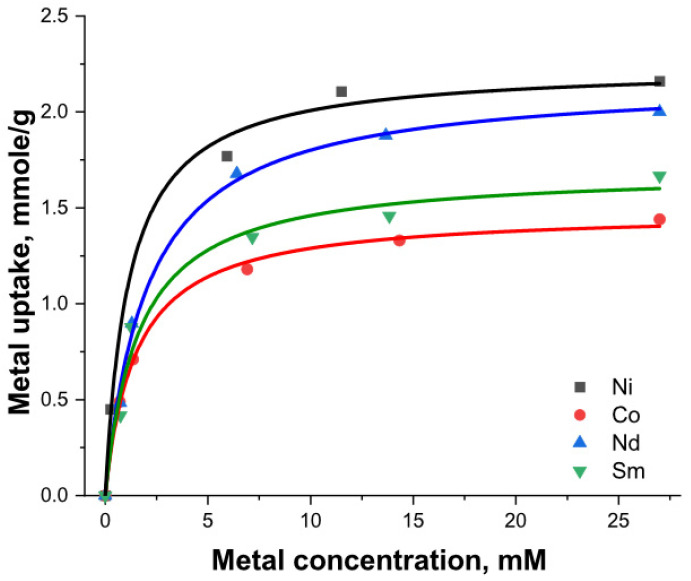
Langmuir isotherms for MMSN-DTPAs.

**Figure 9 nanomaterials-13-03155-f009:**
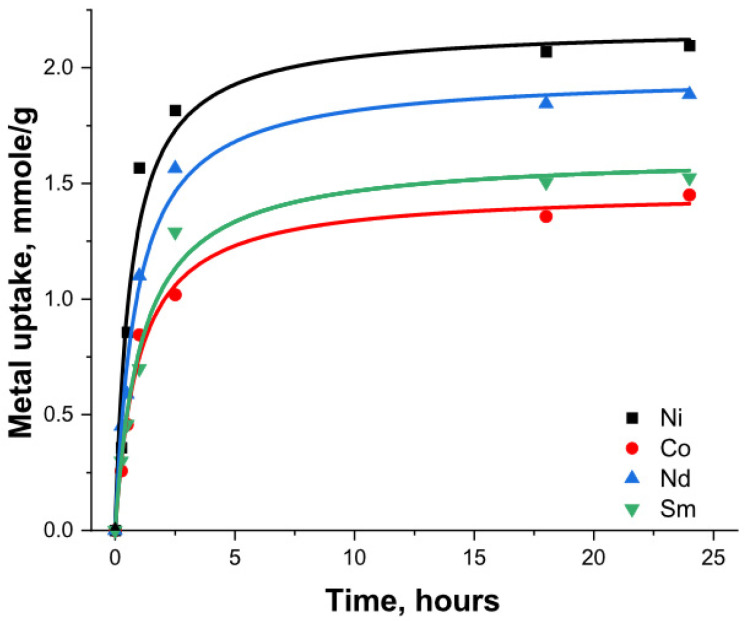
Adsorption kinetics of heavy metals with MMSN-DTPAs.

**Figure 10 nanomaterials-13-03155-f010:**
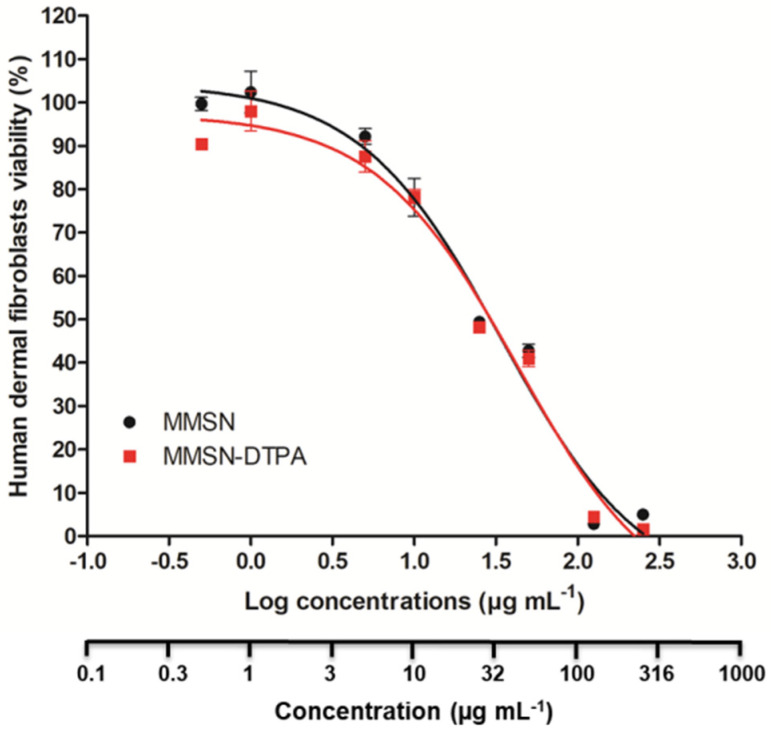
Dose-response curve of healthy human dermal fibroblasts treated with different concentrations of MMSNs or MMSN-DTPAs for 72 h. Data are presented as mean ± SEM of three independent experiments.

**Figure 11 nanomaterials-13-03155-f011:**
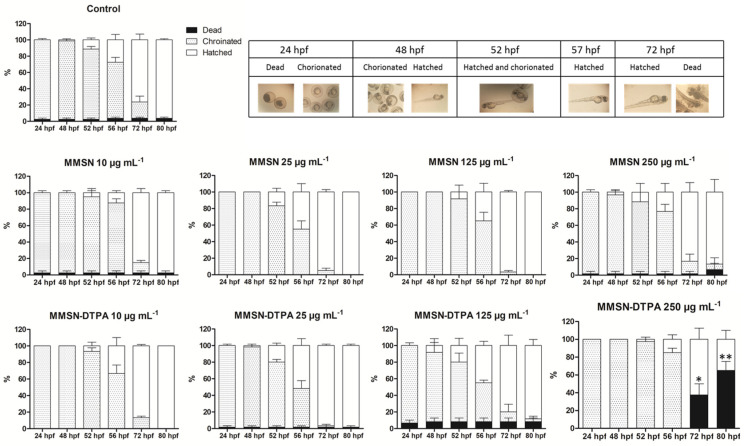
Toxicological study of different concentrations of MMSNs or MMSN-DTPAs on zebrafish embryos’ development at different time points. Data are presented as mean ± SEM of three independent experiments. Images were acquired using Olympus MVX10 stereomicroscope at 2.5 magnification. Dead embryos in the test groups are statistically different from dead embryos in the control group, and the level of significance was defined as * *p* < 0.05 and ** *p* < 0.005.

**Figure 12 nanomaterials-13-03155-f012:**
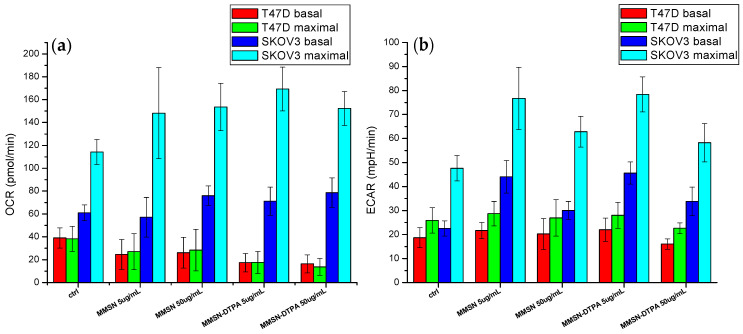
Metabolic measurements of MMSNs and MMSN-DTPAs incubated with T4TD and SKOV3 cells. Oxygen consumption rate measurement (**a**) and extracellular acidification rate measurements (**b**). ctrl means control.

**Table 1 nanomaterials-13-03155-t001:** Average maximum adsorption capacity for each magnetic nanoadsorbent and heavy metal (mmol/g).

Sample	Ni	Co	Nd	Sm
MMSNs	1.00	1.00	0.88	0.85
MMSN-DTPAs	2.16	1.44	2.00	1.66

**Table 2 nanomaterials-13-03155-t002:** List of parameters obtained from Langmuir nonlinear isotherm.

Metal Ion	Ce (mM)	Qmax (mmol/g)	R^2^
Ni	24.8	2.23 (130.9 mg/g)	0.99
Co	25.6	1.48 (87.17 mg/g)	0.99
Nd	25.0	2.16 (311.5 mg/g)	0.98
Sm	25.3	1.69 (254.1 mg/g)	0.99

**Table 3 nanomaterials-13-03155-t003:** Comparison of maximum monolayer adsorption capacities of different adsorbents towards REEs and LTMs, obtained from Langmuir isotherm fitting.

Adsorbent	Metal	Qmax (mg/g)	References
DTPA–chitosan biopolymers	Ni (II)	64.139	[28]
EDTA–chitosan	Ni (II)	77.073	[28]
(N-(2-Aminoethyl)-3-aminopropyltrimethoxysilane SiO_2_ nanoparticles	Ni (II)	106.830	[29]
DTPA–chitosan biopolymers	Co (II)	52.866	[1,28]
EDTA–chitosan	Co (II)	65.466	[28]
(N-(2-Aminoethyl)-3-aminopropyltrimethoxysilane SiO_2_ nanoparticles	Co (II)	111.910	[29]
DTPA–chitosan biopolymers	Nd (III)	74.000	[30]
(N-(2-Aminoethyl)-3-aminopropyltrimethoxysilane SiO_2_ nanoparticles	Nd (III)	161.550	[29]
DTPA-SiO_2_ nano and microparticles	Nd (III)	132.700	[31]
phosphonic acid magnetic nanoparticles	Sm (III)	55.630	[32]
(N-(2-Aminoethyl)-3-aminopropyltrimethoxysilane SiO_2_ nanoparticles	Sm (III)	196.970	[29]
DTPA–chitosan biopolymers	Ni (II)	64.139	[28]
EDTA–chitosan	Ni (II)	77.073	[28]
(N-(2-Aminoethyl)-3-aminopropyltrimethoxysilane SiO_2_ nanoparticles	Ni (II)	106.830	[29]

**Table 4 nanomaterials-13-03155-t004:** Selectivity of MMSNs and MMSN-DTPAs in relation to separation of cations.

Sample	Ni:Nd	Co:Sm	Ni:Co
MMSNs			1:1
MMSN-DTPAs	12:1	1:2.4	9.4:1

## Data Availability

Data availability from the authors.

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
