# Peer review of "Upscale Synthesis of Magnetic Mesoporous Silica Nanoparticles and Application to Metal Ion Separation: Nanosafety Evaluation"

_nanomaterials, 2023, doi:10.3390/nano13243155_

Round 1
Reviewer 1 Report
Comments and Suggestions for Authors
In this study, the researchers focused on synthesizing core-shell magnetic mesoporous nanoparticles (MMSN) at a gram scale, which is a challenging task compared to the conventional 100-250 mg scale. They successfully addressed this challenge through a meticulous phase transfer process. Iron oxide core nanoparticles (IO) were initially synthesized using a thermal decomposition procedure in oleic acid, followed by a phase transfer from chloroform to water. This allowed them to wrap the IO with a mesoporous silica shell using the sol-gel method. The results were impressive, as they achieved a homogeneous dispersion of MMSN at the gram scale. The MMSN was then functionalized with DTPA, demonstrating a strong affinity for removing and separating metal ions, particularly Ni ions. Importantly, toxicity evaluations were conducted in cells, zebrafish, and through seahorse metabolic assays, and the MMSN were found to be non-toxic, even at relatively high concentrations of up to 125 μg/mL. In conclusion, this study presents a successful method for preparing MMSN at a larger scale, making them highly efficient for metal ion removal and separation. The added bonus is that these nanoparticles exhibited no toxicity concerns, even at significant concentrations, making them promising for various practical applications in the field of nanotechnology and environmental science. Because of the abovementioned fact, I strongly recommend this paper for publication in the Nanomaterials journal in its current form (without changes).
Author Response
We warmly thank the reviewer for his positive comments. Some corrections have however been carried out.
Reviewer 2 Report
Comments and Suggestions for Authors
Review on the manuscript entitled “Upscale synthesis of magnetic mesoporous silica nanoparticles and application to metal ion separation, nanosafety evaluation” by Jean-Olivier Durand et al.
I recommend the authors to carefully read all the manuscript and reply to the queries presented below.
Minor
In the abstract – should be removed all the subsection – background, methods…
Line 20 - magnetic mesoporous nanoparticles (MMSN), line 38 Magnetic Mesoporous Silica Nanoparticles (MMSN)
Please note the references before the end of the sentence.
Please carefully check the spelling and the punctuation marks
Major
Introduction should include: the previous utilization of MMSN (or variations) in the removal of various ions, as well as their use in toxicity assessments.
The Preparation of MMSN and MMSN-DTPA should be presented in the experimental part not in the Results and discussion section.
The FTIR spectra should be revised – Transmittance(%) vs Wavenumber(1/cm); all the bands should be allocated
The zeta potential and TGA analyses are not discussed.
The adsorption study should include the equilibrium data – For Langmuir: Ce (mg/ l), qe (mg/g) the R2 coefficient. These results should be discussed comparative to the existing data in the literature.
The in vitro/embryo study is well presented in terms of results, but again the discussion part is missing.
I do not understand the double-use of the particles in the same article; authors should consider dedicating separate articles for the utilizations (adsorbent and in vitro assessment).
The conclusion should only underline the findings and not repeat the results.
The number of references should be higher.
Comments on the Quality of English LanguageMinor editing is required
Author Response
We thank the reviewer for his comments allowing improving the quality oif our manuscript.
Q: In the abstract – should be removed all the subsection – background, methods…
A: The subsections were removed.
Q: Line 20 - magnetic mesoporous nanoparticles (MMSN), line 38 Magnetic Mesoporous Silica Nanoparticles (MMSN)
A: The abbreviation was corrected.
Q: Please note the references before the end of the sentence.
A: The references labels were corrected.
Q: Please carefully check the spelling and the punctuation marks
A: The article was carefully checked
Major
Q: Introduction should include: the previous utilization of MMSN (or variations) in the removal of various ions, as well as their use in toxicity assessments.
A: More references were added, and briefly discussed.
Q: The Preparation of MMSN and MMSN-DTPA should be presented in the experimental part not in the Results and discussion section.
A: The preparation of MMSN, based on upscaling, was a very difficult task. We believe the difficulties of upscaling should be highlighted in results and discussion part.
Q: The FTIR spectra should be revised – Transmittance(%) vs Wavenumber(1/cm); all the bands should be allocated
A: The FTIR spectra were revised and the bands allocated.
Q: The zeta potential and TGA analyses are not discussed.
A: Discussion and reference were added.
Q: The adsorption study should include the equilibrium data – For Langmuir: Ce (mg/ l), qe (mg/g) the R2 coefficient. These results should be discussed comparative to the existing data in the literature.
A: The R2 coefficient was added and comparison with data from literature explained.
Q: The in vitro/embryo study is well presented in terms of results, but again the discussion part is missing.
A: Discussion was added with reference 33
Q: I do not understand the double-use of the particles in the same article; authors should consider dedicating separate articles for the utilizations (adsorbent and in vitro assessment).
A: The analysis of the safety of the nanoparticles is an important issue, which should not be neglected. Therefore, it is important not to separate utilization and safety analysis.
Q: The conclusion should only underline the findings and not repeat the results.
A: The conclusion was modified and improved.
Q: The number of references should be higher.
A: The number of references was increased and described in the introduction
Reviewer 3 Report
Comments and Suggestions for Authors
The authors present an article that in my opinion can be of the interest of the Nanomaterials readership but requires revision before publication.
The authors present the article as mainly an upscale synthesis of previously synthesized magnetic nanoparticles to gram scale. However in section 3, subsection “synthesis of MMSN”, “Functionalisation of MMSN with DTPA”, or in the manuscript text in the discussion, I can’t find the exact amounts of reagents that were used in the synthesis and the amount of the nanoparticles the authors obtained.
The meaning of all abbreviations should be stated on their first appearance.
I was unable to find / review the supplementary material / figure S1.
Line 27, 114: The authors mention glassware as a critical parameter but not describe how, only at the conclusions there is a mention to the use of larger neck glassware
Line 64, 212: seahorse metabolic assays/measurements: Since the authors are not analysing seahorses (animals) but using a specific technique/instrument (Agilent Seahorse XF analyser) this should be stated clearly for people who might not be familiarized with the technique.
Lines 163-165 (selectivity tests): On my view all combinations should have been tested (Ni:Co, Ni:Sm, Ni:Nd, Co:Sm, Co:Nd, Sm:Nd), or at least all combinations in one of the ions, such as Ni or Co. Perhaps other common metal ions should also have been tested.
Line 187: Figure in 1I – Do the authors mean 11?
Lines 267-273 – Materials and methods: The purity of all reagents should be stated.
Line 377: The equation must include the left hand side, e.g. %(viable cells) = Abtest/Abcontrol *100
Line 409/410: seahorse, RPMI medium – Do the authors mean: Agilent Seahorse XF RPMI Medium ?
Lines 436-438: The header before “Conceptualization” should be removed!
I wish the authors all the best in the continuation of their work.
Comments on the Quality of English LanguageI believe the English language of the article is reasonably correct (I'm not a native speaker)
Author Response
The authors present an article that in my opinion can be of the interest of the Nanomaterials readership but requires revision before publication. The authors present the article as mainly an upscale synthesis of previously synthesized magnetic nanoparticles to gram scale.
A: We thank the reviewer ofr his comments allowing improving the quality of the manuscript. The corrections are highlighted in red.
Q: However in section 3, subsection “synthesis of MMSN”, “Functionalisation of MMSN with DTPA”, or in the manuscript text in the discussion, I can’t find the exact amounts of reagents that were used in the synthesis and the amount of the nanoparticles the authors obtained.
A: The amounts were precised and the yield roughly calculated based on TEOS consomption.
Q: The meaning of all abbreviations should be stated on their first appearance.
A: The abbreviations were explained
Q: I was unable to find / review the supplementary material / figure S1. Line 27, 114:
A: The mistake was corrected there is no supplementary material
Q: The authors mention glassware as a critical parameter but not describe how, only at the conclusions there is a mention to the use of larger neck glassware
A: The glassware was precised in materials and methods.
Q: Line 64, 212: seahorse metabolic assays/measurements: Since the authors are not analysing seahorses (animals) but using a specific technique/instrument (Agilent Seahorse XF analyser) this should be stated clearly for people who might not be familiarized with the technique.
A: The metabolic assay was precised
Q: Lines 163-165 (selectivity tests): On my view all combinations should have been tested (Ni:Co, Ni:Sm, Ni:Nd, Co:Sm, Co:Nd, Sm:Nd), or at least all combinations in one of the ions, such as Ni or Co. Perhaps other common metal ions should also have been tested.
A: We have added a table comparing MMSN-DTPA with other systems from the literature, for the maximum adsorption of late transition metals and rare earth elements. This table shows that MMSN-DTPA is competitive towards the literature. We therefore believe studying all the combinations is not necessary.
Q: Line 187: Figure in 1I – Do the authors mean 11? Lines 267-273 – Materials and methods: The purity of all reagents should be stated. Line 377: The equation must include the left hand side, e.g. %(viable cells) = Ab_test/Ab_control *100 Line 409/410: seahorse, RPMI medium – Do the authors mean: Agilent Seahorse XF RPMI Medium ? Lines 436-438: The header before “Conceptualization” should be removed! I wish the authors all the best in the continuation of their work.
A: The corrections were performed, Roswell Park Memorial Institute medium (RPMI)
Reviewer 4 Report
Comments and Suggestions for Authors
The paper of J. O. Durand et al. is an original study making a step forward in the MMSM nanostructures large-scale synthesis, and the findings of the authors are substantial. The MMSN preparation was improved, resulting in nanoparticles with better metal-scavenger properties, in nanosafety limits and the resulted core-shell nanoparticles were characterized through up-to-date methods. To ensure a better applicability of the results in further studies, some aspects have to be clarified in the paper.
Abstract: DTPA and α-FeO meaning should be explained in abstract. α-FeO appeared later, in the conclusions as αFeO, and nowhere in the results or experimental section is no data about this compound, instead other terms refer to the same NP’s.
Results: Authors should explain why Ni2+, Co2+, Sm3+ and Nd3+ ions were chosen for the experiment, for the readers who are less familiar with ion extraction and separation techniques. How they improved the nanostructures in order to obtain better affinity for these ions?
The authors mentioned in the introduction and conclusions that three parameters made the difference in synthesis: temperature, time and glassware. In the results section the parameters are described, but they should be enlightened in detail in the methods section as well (for example the volume, shape of the used glassware). Which are the main differences in compare with previous synthesis methods; examples with citations.
Figure 1: more data about the general term: “ligand” should be stated
Figure 4: explain HRTEM abbreviation. This term appears nowhere else in the text, than why it is important to depict it?
Page 4: “Zeta analyses” needs be edited.
Page 11, Material and Methods: providers of all reagents, equipment manufacturers should be mentioned rigorously, comprising the exact name of the company, city, country.
Page 12, What was the yield of the MMSN-DTPA synthesis? Which parameters were optimized by the authors, set against previous papers?
Page 13, paragraph 2: MNP’s were the same as MMSN described in the previous sections, or other nanoparticles? If the same, please the regularize the abbreviations, if different, please insert the source/procedure.
MION’s were described in the methods section, but they were mentioned nowhere in the Results section. Please check the correspondence between the term used in different chapters of the manuscript.
Page 13- 15: Cell cultures, cytotoxicity, in vivo study, metabolic assay
Manufacturer, city, country for cell culture media, supplements, MTT, DMSO, consumables, incubator, microscope or other essential equipment is missing. Same for the cell bank and zebrafish source. The microplate reader used for absorbance measurements was not mentioned. MMDN complete name should be inserted. It was not mentioned if the in vitro and zebrafish embryo experiment were performed in triplicates (as mentioned one time, in Figure 10 capction), how many independent experiments were done. Please explain 80 hpf.
The source of SK-OV-3 and T47D cells was not specified. The oxygen consumption test which employ Agilent Seahorse XF96 device should be described briefly. “Two cell groups” is consisting of how many wells, batches, duplicates or other ways to group the samples? Please rewrite: “incubated in unbuffered, SEAHORSE, RPMI medium”. FCCP abbreviation was not explained (in Results section, where the term was first mentioned).
All over the text, some typos need to be revised.
Author Response
The paper of J. O. Durand et al. is an original study making a step forward in the MMSM nanostructures large-scale synthesis, and the findings of the authors are substantial. The MMSN preparation was improved, resulting in nanoparticles with better metal-scavenger properties, in nanosafety limits and the resulted core-shell nanoparticles were characterized through up-to-date methods. To ensure a better applicability of the results in further studies, some aspects have to be clarified in the paper.
A: We thank the reviewer who allowed to improve the quality of the manuscript. The corrections are highlighted in red in the uploaded manuscript.
Q: Abstract: DTPA and α-FeO meaning should be explained in abstract. α-FeO appeared later, in the conclusions as αFeO, and nowhere in the results or experimental section is no data about this compound, instead other terms refer to the same NP’s.
A: DTPA was explained and the mistake α-FeO was replaced with α-FeO(OH)
Q: Results: Authors should explain why Ni2+, Co2+, Sm3+ and Nd3+ ions were chosen for the experiment, for the readers who are less familiar with ion extraction and separation techniques. How they improved the nanostructures in order to obtain better affinity for these ions?
A: REEs are key components of green energy and high-tech growth industries. For example There are about 320 kg of rare earth elements (REE) in the magnet of a wind turbine. Separating elements from outdated systems is therefore crucial for recycling REE. Functionalizing MMSN with DTPA allowed to improve affinity for Ni2+.
Q: The authors mentioned in the introduction and conclusions that three parameters made the difference in synthesis: temperature, time and glassware. In the results section the parameters are described, but they should be enlightened in detail in the methods section as well (for example the volume, shape of the used glassware). Which are the main differences in compare with previous synthesis methods; examples with citations.
A: The main differences were highlighted in the text and compared with citations.
Q: Figure 1: more data about the general term: “ligand” should be stated
A: Ligand was replaced with DTPA.
Q: Figure 4: explain HRTEM abbreviation. This term appears nowhere else in the text, than why it is important to depict it?
A: HRTEM: High-resolution transmission electron microscopy
Q: Page 4: “Zeta analyses” needs be edited.
A: Zeta analyses were explained
Q: Page 11, Material and Methods: providers of all reagents, equipment manufacturers should be mentioned rigorously, comprising the exact name of the company, city, country.
A: The Material and Methods were completed.
Q: Page 12, What was the yield of the MMSN-DTPA synthesis? Which parameters were optimized by the authors, set against previous papers?
A: The parameters were highlighted in the text and in materials and methods and the yield roughly calculated based on TEOS consomption.
Q: Page 13, paragraph 2: MNP’s were the same as MMSN described in the previous sections, or other nanoparticles? If the same, please the regularize the abbreviations, if different, please insert the source/procedure.
A: MNP was replaced by MMSN
Q: MION’s were described in the methods section, but they were mentioned nowhere in the Results section. Please check the correspondence between the term used in different chapters of the manuscript.
A: MION’s was replaced with IO
Q: Page 13- 15: Cell cultures, cytotoxicity, in vivo study, metabolic assay
A: Manufacturer, city, country for cell culture media, supplements, MTT, DMSO, consumables, incubator, microscope or other essential equipment is missing. Same for the cell bank and zebrafish source. The microplate reader used for absorbance measurements was not mentioned. MMDN complete name should be inserted. It was not mentioned if the in vitro and zebrafish embryo experiment were performed in triplicates (as mentioned one time, in Figure 10 capction), how many independent experiments were done. Please explain 80 hpf.
Q: Comment of the reviewer has been taken into account and the needed corrections were done. Experiments were repeated 3 times. 80 “hpf” abbreviation was already mentioned in the same paragraph which means hours post fertilization (hpf).
Q: The source of SK-OV-3 and T47D cells was not specified. The oxygen consumption test which employ Agilent Seahorse XF96 device should be described briefly. “Two cell groups” is consisting of how many wells, batches, duplicates or other ways to group the samples? Please rewrite: “incubated in unbuffered, SEAHORSE, RPMI medium”. FCCP abbreviation was not explained (in Results section, where the term was first mentioned).
A: The corrections were performed.
Q: All over the text, some typos need to be revised.
A: The typo were corrected.
Round 2
Reviewer 2 Report
Comments and Suggestions for Authors
The manuscript is improved and can be considered for publication.
Comments on the Quality of English LanguagePlease carefully check your manuscript, some typos need to be revised.
Author Response
We thank the reviewer for his positive comments,
Typos were corrected
Reviewer 3 Report
Comments and Suggestions for Authors
In my opinion, in general the authors have answered adequately to my remarks and made an effort to improve the manuscript.
A: We have added a table comparing MMSN-DTPA with other systems from the literature, for the maximum adsorption of late transition metals and rare earth elements. This table shows that MMSN-DTPA is competitive towards the literature. We therefore believe studying all the combinations is not necessary.
R: In my opinion adding the other combinations would improve the work but I understand it is not easy to carry on more tests in this phase of the work and I believe this is not crucial for the paper. In my opinion the table the authors added enriches the manuscript.
Now line 211: Figure in 1I - Please check if there is some mispelling here
I wish the authors all the best in the continuation of their work.
Author Response
We thank the reviewer for his positive comments,
The typos were corrected.